# Adverse fetal birth outcomes and its associated factors among mothers with premature rupture of membrane in Amhara region, Ethiopia

**Abebe Abrha Alene** *, **Endalkachew Worku Mengesha, Gizachew Worku Dagnew**

Department of Reproductive Health and Population Studies, School of Public Health, College of Medicine and Health Science, Bahir Dar University, Bahir Dar, Ethiopia

* abrhaabebe778@gmail.com

## Abstract

### Background

Adverse birth outcomes are the leading cause of neonatal mortality worldwide. Ethiopia is one of the countries struggling to reduce neonatal mortality through different strategies, but neonatal mortality remains high for many reasons. Despite adverse birth outcomes being a public health problem in Ethiopia, the contribution of Premature rupture of the membrane to the adverse fetal birth outcome is neglected and not well explained in our country. This study aims to assess fetal birth outcomes and associated factors among mothers with all types of PROM at Specialized Hospitals in Amhara Region, Ethiopia.

### Methods

A facility-based cross-sectional study design was applied among 538 mothers with premature rapture of the membrane at Amhara region specialized hospitals. A simple random sampling technique was employed to select the medical charts diagnosed with all types of PROM and giving birth in the hospital within the period from July 8, 2019, to July 7, 2021. The data was collected using a checklist, entered into EPI Data version 3.1, and analyzed using SPSS version 23. A binary logistic regression model was used to see the association between independent and dependent variables. A P-value <0.05 was used to declare the statistical significance. The AOR with 95% CI was used to measure the strength of the association.

### Result

Adverse birth outcome among all types of Premature rupture of membrane mothers was 33.1% [95% CI 29.2–37.2]. Rural residents [AOR = 2.94, 95% CI:1.73–4.97], have a history of urinary tract infection [AOR = 6.87, 95% CI: 2.77–17.01], anemia [AOR = 7.51, 95% CI: 2.88–19.62], previous history of adverse birth outcome [AOR = 3.54, 95% CI: 1.32–9.47] and less than two years interpregnancy interval [AOR = 6.07, 95% CI: 2.49–14.77] were positively associated with adverse birth outcome compared to their counterparts.

**Data Availability Statement:** All relevant data are within the paper.

**Funding:** The author(s) received no specific funding for this work.

**Competing interests:** The authors have declared that no competing interests exist.

**Abbreviations:** ANC, Antenatal Care; ABO, Adverse birth outcome; AOR, Adjusted Odds Ratio; APH, Ante-Partum Hemorrhage; CI, Confidence Interval; CS, Caesarian Section; DM, Diabetes Mellitus; GA, Gestational Age; NRFHR, None Reassuring Fetal Heart rate; PNA, Perinatal Asphyxia; PROM, Premature Rupture of Membrane; RD, Respiratory Distress Syndrome; RH, Reproductive Health; SVD, Spontaneous Vaginal Delivery; UTI, Urinary Tract Infection; WHO, World Health Organization.

## Conclusion

The adverse birth outcome was high in the Amhara region as compared to the World Health Organization's estimated figure and target; the target is less than 15%. History of the previous adverse birth outcome, residence, urinary tract infection, Anemia, and interpregnancy interval had an association with adverse birth outcomes. Therefore, strengthening close follow-up for mothers who had previous adverse birth outcomes, screening and treatment of urinary tract infection, anemia prevention, and maximizing birth interval are recommended for reducing adverse birth outcomes.

## Introduction

Premature rupture of membranes (PROM) is a rupture of the membranes after the 28th week of gestation and before the onset of labor [1]. PROM occurs in 3–8% of all pregnancies [2–4]. This figure reaches around 14% in Ethiopia [5]. PROM has a great contribution to many fetal adverse birth outcomes, like; stillbirth, low birth weight, preterm birth, and congenital anomalies [6, 7].

PROM can cause stillbirth and other fetal adverse birth outcomes due to umbilical cord prolapse and compression. According to the World Health Organization (WHO), Stillbirth is defined as a baby who dies after 28 weeks of pregnancy [8]. Preterm birth and Low birth weight are other adverse birth outcomes of PROM which is defined as babies born alive before 37 weeks of pregnancy and below 2,500 grams respectively [8, 9]. Congenital anomalies are another serious adverse birth outcome characterized by structural or functional anomalies that occur during intrauterine life [8, 10]. Rupture of the membrane can cause pulmonary hypoplasia, and positional deformities of the hands and feet [8, 11, 12].

Adverse birth outcomes are the leading cause of neonatal morbidity and mortality worldwide, mainly in developing countries [7, 13]. Globally, PROM contributed to 20% of stillbirths, 33% of prematurity, 21% of early neonatal death, and 2% of congenital anomalies [7, 13, 14]. Besides, approximately 22% of women experienced unfavorable maternal outcomes due to PROM [15, 16].PROM is one of the complications linked to significant maternal and fetal morbidity and mortality. The magnitude of the fetal adverse birth outcome among PROM varies in different countries. Evidence revealed that premature rupture of the membrane causes 14% to 63% of adverse birth outcomes [17]. Up to 50% of preterm births and 80% of maternalinfections have been associated with PROM [18, 19]. Ethiopia has been implementing different strategies and programs to achieve the neonatal health target set in the global sustainable development goals (SDGs) and the second national Health sectors transformation plan (HSTP-II). Of those, promoting maternal health continuum of care, new-born corners, helping babies breath, Neonatal Intensive Care Units (NICU), community, and facility-based Integrated Management of Neonatal and Childhood Illnesses (IMNCI), programs are now implemented in the country to save the lives of the fetus and neonates. However, Perinatal and neonatal mortality remains high in the country (31 neonatal deaths per 1000 live births), particularly in the Amhara region (33 neonatal deaths per 1000 live births) [20]. PROM-related adverse outcomes take the leading role in this unacceptably high perinatal and neonatal mortality [21].

Although the country failed to achieve the millennium development goals as well as the first national health sector transformation plan of reducing neonatal mortality to 28/1000 live

births, as per the authors' knowledge there is no evidence of the magnitude of adverse fetal birth outcome for the last 5–10 years. Recent evidence would be a good source of knowledge for public health experts to understand the implication of adverse birth outcomes among women with premature rupture of membranes and to visualize the area of intervention for reducing perinatal mortality.

This would also have a great contribution to achieving global and national goals. Therefore, this study aims to assess the fetal adverse birth outcomes and associated factors among PROM mothers who gave birth at specialized hospitals, in the Amhara region, Ethiopia.

## Methods and materials

### Study design and study area

A facility-based cross-sectional study was employed among PROM mothers in Amhara regional state specialized hospitals. Amhara region is the second most populous region in Ethiopia with around 24 million people living in the region. There are 52 primary hospitals, 9 general Hospitals, and 8 specialized hospitals in the Amhara region. Comprehensive specialized hospital covering a population of 3.5–5 million people. The hospitals that give a tertiary level of care including all obstetric emergencies and neonatal intensive care serve as referral centers from general and primary hospitals. The study was conducted in four specialized hospitals: Debre Brihan Specialized Hospital, Debre Tabor Specialized Hospital, Felege Hiwot Specialized Hospital, and Debre Markos Specialized Hospitals which are found in Amhara region, Ethiopia.

### Sample size determination

The sample size was determined using the double population proportion formula using factors associated with adverse birth outcomes. Antenatal care follow-up status was taken in the previous study to determine the current sample [22]. Assumptions; two-sided level significance: 95%, power: 80%, percent with outcome among mothers who had no ANC follow-up:50%, percent with outcome among those who had ANC follow-up:30%, the ratio of sample size: 2, design effect for reducing the error due to multi-stage sampling: 1.5. Finally, the largest sample size was 538, which was the largest sample that determined using factors and single population proportion as well.

### Sampling techniques

Of a total of eight specialized public hospitals in the Amhara region, four of them were selected using the lottery method. The study was conducted on four selected hospitals; Debere Tabor, Debre Markos, Felege Hiwot, and Debre Brihan specialized hospital. The computed sample size was allocated proportionally based on the last two-year PROM cases. The data were collected retrospectively from July 08, 2019, to July 07, 2021. All medical records of pregnant women diagnosed with all types of PROM during this period were included. The medical record were assessed from November 01,2022 to December 30,2022. The mother's medical record was recruited using a computer-generated simple random sampling technique (Fig 1).

### Data collection tool and measurement

A written informed consent letter was obtained for data collection from each hospital on behalf of patients to use patients' medical folder for this study. Four trained hospital matrons collected data from patients' folders by using a data collection checklist. The checklist had four parts; sociodemographic variables, Health service utilization, obstetric history, and maternal

**Fig 1. Sampling procedure and techniques to select mothers with all types of PROM in Amhara region specialized hospitals, Ethiopia, 2022.**

and fetal outcomes. The checklist was prepared by reviewing different related research articles and guidelines [16, 23, 24]. Moreover, the tool was reviewed by four experts from the Amhara Regional Health Bureau and Bahir Dar University. Before finalized, the checklist was cross-checked with client charts to confirm the availability of data.

The outcome variable (Adverse birth outcomes) was declared when the mother with all types of premature rupture of the membrane had one or more of the following; stillbirth, prematurity, low birth weight, and congenital anomalies. Furthermore, Age, residence, Gravidity, History of Abortion, History of previous adverse birth outcome, Duration of PROM, Onset of Labour, Color of liquor, inter-pregnancy interval, Mode of delivery, sex of baby, Urinary tract infection, sexually transmitted disease, HIV/AIDS, DM, Anemia, and ANC service utilization

were collected from medical charts as an independent variable. Those charts that had incomplete information were managed by a random replacement mechanism.

## Data quality assurance

To ensure the quality of the data; before the actual data collection, a pretest was done in Felege Hiwot specialized hospitals on 5% of the current sample size to identify the missing variables on the checklist that are found in the clients' records, as well as to correct the misrepresenting variables in the checklist for the final data collection. Two days of training were given for data collectors and supervisors on the objectives of the study and the overall process of data collection and handling. During the data collection, the supervisors and the principal investigators strictly followed the data collection process, and they provided corrective feedback to data collectors daily. After data collection, appropriate coding and handling of each data were done.

## Data analysis

The collected data were coded and entered using EPI data and exported to SPSS version 23 for analysis. Descriptive analyses such as percentages, frequency distribution, and measures of central tendency were conducted. Then bivariate analyses between dependent and independent variables were performed using bivariate logistic regression. Finally, those variables showing association at a p-value less than 0.2 were considered into multivariable logistic regression analyses to control possible confounding and to identify independent predictor variables of adverse birth outcomes. To declare statistical significance p-value < 0.05 and a 95% confidence interval (CI) were used. Efforts were made to confirm the fulfillment of the major assumption of logistic regression. The absence of multicollinearity was found to be satisfied. The goodness of fit was checked by the Hosmer and Lemeshow model fit-test, P>0.05.

## Results

A total of 538 mothers with all types of PROM participated in the study, This study finding showed that the prevalence of fetal adverse birth outcomes among all types of PROM mothers was 33.1% (95% CI 29.2–37.2). Out of the adverse birth outcomes (60%) of the outcomes were prematurity, (26%) were LBW, (9%) were stillbirth, and (5%) were visible congenital anomalies.

### Socio-demographic profile of the participant

The mean age was 27.00 years with a standard deviation of ± 5.00. More than one-third (38.3%) of mothers were in the age group of 25–29 years and teenage mothers were 40 (7.4%). More than half of the respondents (64.9%) lived in an urban setting. Regarding to region of residency, 509 (94.6%) mothers were from the Amhara region (Table 1).

### Medical and health service utilization factor

Most of the mothers, 518 (96.3%) had at least one ANC follow-up. Regarding the mothers' medical conditions, 108 (20.1%) had a diagnosis of urinary tract infection, 114 (21.2%) had anemia, seven (1.3%) had HIV infection, 10 (1.9%) had a history of other STI, and nine (1.7%) mothers had a history of known DM (Table 2).

### Obstetric factor

Among the study participants, 222 (41.2%) were primigravida. Thirty-nine (7.2%) mothers had a history of previous adverse birth outcomes. Regarding the current birth, 207 (38.4%)

**Table 1. The socio-demographic characteristics of a mother's diagnosis with PROM at Amhara, a region-specialized hospital from July 08, 2019, to July 07, 2021, (n = 538).**

| Variables | Categories | Frequency | Percent (%) |
|---|---|---|---|
| Maternal Age | 15–19 | 40 | 7.4 |
| | 20–24 | 148 | 27.5 |
| | 25–29 | 206 | 38.3 |
| | 30–34 | 90 | 16.7 |
| | 35–39 | 43 | 8 |
| | 40–44 | 11 | 2 |
| Residence | Urban | 349 | 64.9 |
| | Rural | 189 | 34.1 |
| Region of residency | Amhara | 509 | 94.6 |
| | Oromia | 19 | 3.5 |
| | Others | 10 | 1.9 |

mothers had less than two years of birth interval, seven (1.3%) mothers had APH, and 22 (4.1%) mothers had pre-eclampsia/eclampsia. A total of 379 (70.4%) mothers initiated their labor spontaneously and 368 (68.4%) mothers had a spontaneous vaginal delivery (Table 3).

## Factors associated with adverse birth outcome

On bivariate analysis residency, gravidity, birth interval, history of abortion, anemia, UTI, and previous history of adverse birth outcomes were candidate variables for multi-variable analysis at a P-value less than 0.2. On multivariable analysis residency, birth interval, UTI, Anemia, and previous history of abortion were significantly associated with adverse birth outcomes at a P-value less than 0.05.

Multivariable logistic regression analysis showed that mothers who lived in rural areas had higher odds of experiencing adverse birth outcomes as compared to those who lived in an urban area [AOR = 2.94, 95% CI (1.73–4.97)]. Mothers who had urinary tract infections were 7 times more likely to experience adverse birth outcomes than those mothers who didn't have urinary tract infections [AOR = 6.87, 95% CI (2.77–17.01)]. Similarly, mothers with a

**Table 2. The health service and medical characteristics of mother's diagnosis with PROM at Amhara, region specialized hospitals from July 08, 2019, to July 07, 2021, (n = 538).**

| Variables | Categories | Frequency | Percent (%) |
|---|---|---|---|
| ANC follow-up | Yes | 518 | 96.3 |
| | No | 20 | 3.7 |
| HIV status | Positive | 7 | 1.3 |
| | Negative | 531 | 98.7 |
| UTI | Yes | 108 | 20.1 |
| | No | 430 | 79.9 |
| History of STI | Yes | 10 | 1.9 |
| | No | 528 | 98.1 |
| History of DM | Yes | 9 | 1.7 |
| | No | 529 | 98.3 |
| History of previous HTN | Yes | 2 | 0.4 |
| | No | 536 | 96.6 |
| Anemia | Yes | 114 | 21.2 |
| | No | 424 | 78.8 |

**Table 3. The obstetric factors of mother's diagnosis with PROM at Amhara, region specialized hospitals from July 08, 2019, to July 07, 2021, (n = 538).**

| Variables | Categories | Frequency | Percent (%) |
|---|---|---|---|
| Number of pregnancies | Primigravida | 222 | 41.2 |
| | 2–3 | 223 | 41.5 |
| | ≥4 | 93 | 17.3 |
| Interpregnancy interval | < 2years | 207 | 38.4 |
| | ≥ 2years | 109 | 20.2 |
| Antepartum hemorrhage | Yes | 7 | 1.3 |
| | No | 531 | 98.7 |
| Pre-eclampsia/Eclampsia | Yes | 22 | 4.1 |
| | No | 516 | 95.9 |
| Onset of labor | Spontaneous | 379 | 70.4 |
| | Induced | 159 | 29.6 |
| Color of liquor | Clear | 462 | 85.9 |
| | Meconium S. | 65 | 12.1 |
| | Bloodstained | 11 | 2 |
| Mode of delivery | SVD | 368 | 68.4 |
| | C/S | 133 | 24.7 |
| | Instrumental | 37 | 6.9 |
| Duration of PROM to reach the hospital | <12 hrs. | 294 | 54.6 |
| | >12 hrs. | 244 | 55.4 |
| The total duration of PROM to delivery | <24 hrs. | 327 | 60.8 |
| | >24 hrs. | 211 | 39.2 |

hemoglobin level less than 11 gm/dl had odds of developing adverse birth outcomes than those mothers with a hemoglobin level greater or equal to 11 gm/dl [AOR = 7.51, 95% CI (2.88–19.62)]. Mothers who had a history of previous adverse birth outcomes were 4 times more likely to experience adverse birth outcomes compared to their counterparts [AOR = 3.54, 95% CI (1.32–9.47)]. There were higher odds of adverse birth outcomes among women who had less than two years of the inter-pregnancy interval as compared to those who had more than two years [AOR = 6.07, 95% CI (2.49–14.77)] (Table 4).

## Discussion

Premature rupture of the membrane is one of the complications linked to significant maternal and fetal morbidity and mortality. Adverse fetal birth outcomes are the leading cause of neonatal morbidity and mortality worldwide, mainly in developing countries [7, 9].

The main objective of this study was to assess the fetal adverse birth outcome among PROM mothers. This study finding showed that the prevalence of fetal adverse birth outcomes among PROM mothers was 33.1% (95% CI 29.2–37.2). Rural residents [AOR = 2.94, 95% CI:1.73–4.97], have a history of urinary tract infection [AOR = 6.87, 95% CI: 2.77–17.01], anemia [AOR = 7.51, 95% CI: 2.88–19.62], previous history of adverse birth outcome [AOR = 3.54, 95% CI: 1.32–9.47] and less than two years interpregnancy interval [AOR = 6.07, 95% CI: 2.49–14.77] were positively associated with adverse fetal birth outcome.

The prevalence of adverse fetal birth outcomes in this study was higher than the WHO estimation of fetal adverse birth outcomes of 15.5% (7% for most developed countries and 18.5% for developing countries) [9].

The finding of this study was in line with the study conducted at Mizan Aman in the southern part of Ethiopia, 31% [16]. However, the current finding is lower than the study conducted

**Table 4. Multivariable analysis adverse birth outcome and associated factors of mother's diagnosis with PROM at Amhara, region specialized hospitals from July 08, 2019, to July 07, 2021, (n = 538).**

| Variables | | Adverse birth outcome | | COR (95% CI) | AOR (95% CI) |
|---|---|---|---|---|---|
| | | Yes | No | | |
| Residency | Rural | 85 | 104 | 2.25(1.55–3.27) ** | 2.94 (1.73–4.97) * |
| | Urban | 93 | 256 | 1 | 1 |
| UTI | Yes | 79 | 29 | 9.11 (5.63–14.74) ** | 6.87 (2.77–17.01) ** |
| | No | 99 | 331 | 1 | 1 |
| Hemoglobin | <11gm/dl | 94 | 20 | 19.01 (11.11–32.6) ** | 7.51 (2.87–19.62) ** |
| | >11gm/dl | 84 | 340 | 1 | 1 |
| History of Abortion | Yes | 44 | 25 | 4.4 (2.59–7.48) * | 1.59 (0.73–3.43) |
| | No | 134 | 335 | 1 | 1 |
| History of previous adverse birth outcome | Yes | 31 | 8 | 9.27 (4.16–20.66) ** | 3.54 (1.32–9.47) ** |
| | No | 147 | 352 | 1 | 1 |
| Interpregnancy interval | <2years | 113 | 94 | 11.90 (5.87–24.09) ** | 6.07 (2.49–14.77) ** |
| | > 2years | 10 | 99 | 1 | 1 |
| Gravidity | 2–3 | 73 | 150 | 1.51 (1.00–2.29) | 1.19 (0.15–9.49) |
| | ≥4 | 51 | 42 | 3.77 (0.25–0.68) * | 0.9 (0.44–1.81) |
| | Primigravida | 54 | 168 | 1 | 1 |

*p<0.05

**p<0.01

in Egypt, 61.3% [6]. The lower adverse birth outcome in the current study might be a result of the higher number of smokers (10%) and teenage (15%) mothers in a study done in Egypt than in the current study. Smokers and teenage pregnancies are more at risk for fetal adverse birth outcomes [25]. The other reason might also be the difference in study participants; in a study done in Egypt, the study participants were mothers with gestational age below 37 weeks, whereas all types of PROM mothers were included in the current study. Evidence revealed that fetal adverse birth outcome is higher among Mothers who have preterm PROM than mothers who develop PROM after 37 weeks of gestation [26].

On the other hand, the finding of this study was higher than the study findings done in Germany, 25% [27]. The higher magnitude of fetal adverse birth outcomes in the current study might be due to the low accessibility and quality of maternal health services in Ethiopia as compared to Germany and other socioeconomic and knowledge differences might also have contributed to the observed differences.

This study revealed that the odds of fetal adverse birth outcomes among rural study participants were higher than the odds among urban study participants. This finding was supported by the previous studies conducted in Ethiopia [28, 29]. It would have been ideal, in a developing country like Ethiopia maternal healthcare service distribution were not equal in urban and rural residence. Also, the awareness of rural mothers about maternal care services is low compared to urban mothers. Agricultural pesticide and insecticide exposure might be another reason for the higher adverse birth outcomes among women who reside in rural areas. Agricultural pesticide and insecticide exposure can cause 5–9% of adverse birth outcomes [30].

Mothers who have UTIs were positively associated with developing adverse birth outcomes compared to those who didn't have UTIs. This is in line with the studies done in Debre Tabor, Uganda, and India [4, 22, 31]. Urinary tract infections are potential reservoirs for bacteria they pass through the vagina and ascend through the cervical canal to the membranes where they cause localized inflammation. Also, the subsequent prostaglandin production resulting from

localized inflammation leads to occult contractions with increased shearing stress at the cervical os resulting in the premature rupture of the membrane and it may lead to preterm labor (giving birth too early) and low birth weight [31]. As a result, the WHO guideline recommends that all mothers with a positive pregnancy experience be screened for asymptomatic bacteriuria and treated if the results become positive on their ANC contacts [32].

This study also revealed that mothers who had anemia had more odds of developing fetal adverse birth outcomes as compared to the mothers who didn't have it. This finding was concurrent with previous studies done in Indonesia, Bangladesh, and Canada [17, 33, 34]. A lower level of hemoglobin is associated with up to a three-fold increased risk of adverse birth outcomes [35, 36]. The reason could be linked to the effect of anemia on the oxygen-bearing capacity and its transportation to the placental site for the fetus. A low level of hemoglobin or anemia causes a decreased amount of oxygen transported to the tissues, potentially increasing the risk of premature rupture of membranes due to hypoxia in the tissues. Anemia can lead to hypoxia in the tissues, this could cause the fetus not to grow to a healthy weight, may arrive early (preterm birth), or have a low birth weight. Iron deficiency anemia may increase serum concentrations of norepinephrine that cause fetal distress [27].

The mothers with a previous history of adverse birth outcomes were positively associated with current adverse birth outcomes [37]. This finding was in line with a study conducted in Shire town of Northern Ethiopia and Uganda [4, 38]. The previous history of adverse birth outcomes increased the risk of cardiovascular and metabolic diseases and they may have the chance of adverse birth outcomes in the future pregnancy [39]. Preconception care is one of the recommended evidence-based practices to solve the occurrence of similar abnormal birth outcomes in subsequent pregnancies [40].

The finding of our study shows Short inter-pregnancy interval is also found to be a determinant of adverse birth outcomes. The odds of having an adverse birth outcome were higher among mothers having an inter-pregnancy interval of fewer than two years as compared to mothers having a birth interval of more than two years. This finding is in agreement with the other studies conducted in Southern Ethiopia, Uganda, and America [4, 23, 41]. This can happen because of the negative effect of a short inter-pregnancy period on the Mom's body, which does not have enough time to replace nutrient stores before becoming pregnant again, so, obstetrics-related services are important to prevent nutritional deficiencies and other infectious diseases.

This study has some limitations; due to the cross-sectional nature; the study does not show the cause-effect relationships between the predictor and the outcome variables. Further research using primary data is recommended to assess some potential predictor variables missed in the secondary data; maternal occupation, dietary practice, smoking, and alcohol intake habits.

## Conclusion

The prevalence of adverse birth outcomes was high in the Amhara region as compared to the World Health Organization's estimated figure and target. History of the previous adverse birth outcome, residency, Urinary tract infection, Anemia, and inter-pregnancy interval were the predictors for developing adverse birth outcomes among PROM mothers. Therefore, strengthening close follow-up for mothers who had previous adverse birth outcomes, early screening and treatment of urinary tract infection, anemia prevention, and maximizing birth interval is paramount to reducing the adverse birth outcomes among PROM mothers.

## Acknowledgments

We would like thanks to Bahir Dar University College of Medicine and Health Sciences for providing an ethical clearance to conduct this study. Besides, we would like to appreciate all

study hospitals' maternal and child health service unit staff and card room officers. Finally, our gratitude extends to all data collectors and supervisors for their invaluable work.

## Author Contributions

**Conceptualization:** Abebe Abrha Alene, Endalkachew Worku Mengesha, Gizachew Worku Dagnew.

**Data curation:** Abebe Abrha Alene, Gizachew Worku Dagnew.

**Formal analysis:** Abebe Abrha Alene.

**Funding acquisition:** Abebe Abrha Alene.

**Investigation:** Abebe Abrha Alene.

**Methodology:** Abebe Abrha Alene, Endalkachew Worku Mengesha, Gizachew Worku Dagnew.

**Project administration:** Abebe Abrha Alene.

**Resources:** Abebe Abrha Alene.

**Software:** Abebe Abrha Alene, Gizachew Worku Dagnew.

**Supervision:** Endalkachew Worku Mengesha, Gizachew Worku Dagnew.

**Validation:** Abebe Abrha Alene.

**Visualization:** Abebe Abrha Alene.

**Writing – original draft:** Abebe Abrha Alene.

**Writing – review & editing:** Endalkachew Worku Mengesha, Gizachew Worku Dagnew.

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
