## [Decision Letter · Decision Letter 0]

5 Sep 2023

PONE-D-23-07962Adverse Birth Outcomes and Its Associated Factors Among Mothers with Premature Rupture of Membrane in Amhara Region, Ethiopia:PLOS ONE

Dear Dr. Alene,

Thank you for submitting your manuscript to PLOS ONE. After careful consideration, we feel that it has merit but does not fully meet PLOS ONE’s publication criteria as it currently stands. Therefore, we invite you to submit a revised version of the manuscript that addresses the points raised during the review process. 

We look forward to receiving your revised manuscript.

Kind regards,

Kahsu Gebrekirstos Gebrekidan

Academic Editor

PLOS ONE

https://juniperpublishers.com/jgwh/pdf/JGWH.MS.ID.555920.pdf

https://www.scirp.org/

https://journals.indexcopernicus.com/api/file/viewByFileId/393307.pdf

https://www.belitungraya.org/BRP/public/journals/1/BNJ%20Vol%204%283%29-Full.pdf.

In your revision ensure you cite all your sources (including your own works), and quote or rephrase any duplicated text outside the methods section. Further consideration is dependent on these concerns being addressed.

Additional Editor Comments:

Please address the comments forwarded from reviewers.

Reviewers' comments:

Reviewer's Responses to Questions

**Comments to the Author**

1. Is the manuscript technically sound, and do the data support the conclusions?

Reviewer #1: No

Reviewer #2: Partly

2. Has the statistical analysis been performed appropriately and rigorously? 

Reviewer #1: No

Reviewer #2: Yes

3. Have the authors made all data underlying the findings in their manuscript fully available?

Reviewer #1: Yes

Reviewer #2: Yes

4. Is the manuscript presented in an intelligible fashion and written in standard English?

Reviewer #1: Yes

Reviewer #2: No

5. Review Comments to the Author

Reviewer #1: General Comments

There is no novelty in this study because most of factors detailed in this manuscript have been reported repeatedly within various literatures and standard text books. In addition the study design and the method of data collection used are not ideal for such type of studies.

Title

Lacks clarity. It is not clear whether it is maternal or neonatal adverse outcome. In addition it would have been good if the type of PROM were specified.

Introduction

Clearly put, the magnitude of the problem, efforts done to reduce the impact (if any), the identified gaps of those efforts and why your study is particularly useful among others,

Methods and Materials

I am not sure there exits the so called A retrospective facility-based cross-sectional. How can a study design can be a retrospective and cross sectional at the same time? It would have been better if such studies were conducted prospectively.

Be consistent with the numbers of Comprehensive specialized hospital in the region

Include Schematic presentation of sampling procedure

The way the outcome variable declared was not conventional. Did you confident enough that stillbirth, prematurity, low birth weight, and congenital anomalies are adverse birth outcomes solely due to PROM??

It would have been good if validated tool were used for data collection

No mechanism mentioned regarding encountering incomplete medical records

The covariates (independent variables) are few in number, many intrapartum and immediate postpartum variables with potential effect on the outcome variable are missed

The reason why Logistic regression was selected for analysis is not justified using model fitness and other relevant tests

Result

Non respondents are not expected from record review

The result of the first objective is not separately presented

Discussion

Most of the literatures used for discussion are not comparable with your study

Reviewer #2: The authors need to arrange introduction section. Join the paragraphs and rewrite some paraphrase so, the readers will understand what you wanted to convey from messages. The authors should make sure that missing references are written within the introduction section. They also need to make sure to clearly mention the significant of your study. Also, please check grammar to make sure it is more readably for the readers.

6. PLOS authors have the option to publish the peer review history of their article (what does this mean?). If published, this will include your full peer review and any attached files.

Reviewer #1: **Yes: **Mihretu Molla Enyew

Reviewer #2: **Yes: **Zalikha Al-Marzouqi

---

## [Author Response · Author response to Decision Letter 0]

24 Oct 2023

Thank you for your valuable comments!

---

## [Decision Letter · Decision Letter 1]

30 Nov 2023

PONE-D-23-07962R1Adverse Fetal Birth Outcomes and Its Associated Factors Among Mothers with Premature Rupture of Membrane in Amhara Region, Ethiopia.PLOS ONE

Dear Dr. Abebe,

Thank you for submitting your manuscript to PLOS ONE. After careful consideration, we feel that it has merit but does not fully meet PLOS ONE’s publication criteria as it currently stands. Therefore, we invite you to submit a revised version of the manuscript that addresses the points raised during the review process. Please submit your revised manuscript by Jan 14 2024 11:59PM. If you will need more time than this to complete your revisions, please reply to this message or contact the journal office at plosone@plos.org. Please include the following items when submitting your revised manuscript:A rebuttal letter that responds to each point raised by the academic editor and reviewer(s). You should upload this letter as a separate file labeled 'Response to Reviewers'.A marked-up copy of your manuscript that highlights changes made to the original version. You should upload this as a separate file labeled 'Revised Manuscript with Track Changes'.An unmarked version of your revised paper without tracked changes. You should upload this as a separate file labeled 'Manuscript'.If applicable, we recommend that you deposit your laboratory protocols in protocols.io to enhance the reproducibility of your results. Protocols.io assigns your protocol its own identifier (DOI) so that it can be cited independently in the future. For instructions see: https://journals.plos.org/plosone/s/submission-guidelines#loc-laboratory-protocols. Additionally, PLOS ONE offers an option for publishing peer-reviewed Lab Protocol articles, which describe protocols hosted on protocols.io. Read more information on sharing protocols at https://plos.org/protocols?utm_medium=editorial-email&utm_source=authorletters&utm_campaign=protocols.

We look forward to receiving your revised manuscript.

Kind regards,

Kahsu Gebrekidan

Academic Editor

PLOS ONE

Journal Requirements:

Reviewers' comments:

Reviewer's Responses to Questions

**Comments to the Author**

1. If the authors have adequately addressed your comments raised in a previous round of review and you feel that this manuscript is now acceptable for publication, you may indicate that here to bypass the “Comments to the Author” section, enter your conflict of interest statement in the “Confidential to Editor” section, and submit your "Accept" recommendation.

Reviewer #2: All comments have been addressed

Reviewer #3: (No Response)

2. Is the manuscript technically sound, and do the data support the conclusions?

Reviewer #2: Yes

Reviewer #3: Yes

3. Has the statistical analysis been performed appropriately and rigorously? 

Reviewer #2: I Don't Know

Reviewer #3: Yes

4. Have the authors made all data underlying the findings in their manuscript fully available?

Reviewer #2: Yes

Reviewer #3: Yes

5. Is the manuscript presented in an intelligible fashion and written in standard English?

Reviewer #2: Yes

Reviewer #3: No

6. Review Comments to the Author

Reviewer #2: the authors addressed all the comments and they changed according to the comments provided to them. I wish all the best for the authors to publish the study

Reviewer #3: 1. Still the availability of data from client charts is under question since it is secondary data. Since the documentation of client charts in Ethiopia is knowingly poor. Is ethnicity documented in the client chart? Is only replacing mechanism can reduce the bias due to incomplete information?

2. Why pre-test in the same hospital?

3. The discussion section needed some rewriting by using good research discussion section writing steps: - Step 1: Summarize your key findings. Start this section by reiterating your research problem and concisely summarizing your major findings. ...Step 2: Give your interpretations. ...Step 3: Discuss the implications. ...Step 4: Acknowledge the limitations. ...Step 5: Share your recommendations.

4. The manuscript needs some English grammar corrections.

7. PLOS authors have the option to publish the peer review history of their article (what does this mean?). If published, this will include your full peer review and any attached files.

Reviewer #2: **Yes: **Zalikha Al Marzouqi

Reviewer #3: No

---

## [Author Response · Author response to Decision Letter 1]

15 Dec 2023

Thank you for your valuable comment! without your comment we can't produce this amazing article. Thank you once again!

---

## [Decision Letter · Decision Letter 2]

23 Jan 2024

Adverse Fetal Birth Outcomes and Its Associated Factors Among Mothers with Premature Rupture of Membrane in Amhara Region, Ethiopia.

PONE-D-23-07962R2

Dear Mr Abebe,

We’re pleased to inform you that your manuscript has been judged scientifically suitable for publication and will be formally accepted for publication once it meets all outstanding technical requirements.

Kind regards,

Kahsu Gebrekidan

Academic Editor

PLOS ONE

Additional Editor Comments (optional):

Reviewers' comments:

Reviewer's Responses to Questions

**Comments to the Author**

1. If the authors have adequately addressed your comments raised in a previous round of review and you feel that this manuscript is now acceptable for publication, you may indicate that here to bypass the “Comments to the Author” section, enter your conflict of interest statement in the “Confidential to Editor” section, and submit your "Accept" recommendation.

Reviewer #3: All comments have been addressed

2. Is the manuscript technically sound, and do the data support the conclusions?

Reviewer #3: Yes

3. Has the statistical analysis been performed appropriately and rigorously? 

Reviewer #3: Yes

4. Have the authors made all data underlying the findings in their manuscript fully available?

Reviewer #3: Yes

5. Is the manuscript presented in an intelligible fashion and written in standard English?

Reviewer #3: Yes

6. Review Comments to the Author

Reviewer #3: There are no other concerns. The authors have adequately addressed my comments raised in a previous round of review

7. PLOS authors have the option to publish the peer review history of their article (what does this mean?). If published, this will include your full peer review and any attached files.

Reviewer #3: No

---

## [Editor Report · Acceptance letter]

21 Mar 2024

PONE-D-23-07962R2 

PLOS ONE

Dear Dr. Alene, 

I'm pleased to inform you that your manuscript has been deemed suitable for publication in PLOS ONE. Congratulations! Your manuscript is now being handed over to our production team.

Kind regards, 

on behalf of

Dr. Kahsu Gebrekidan 

Academic Editor

PLOS ONE